# Ultraprotective Ventilation via ECCO2R in Three Patients Presenting an Air Leak: Is ECCO2R Effective?

**DOI:** 10.3390/jpm13071081

**Published:** 2023-06-29

**Authors:** Carolina Ferrer Gómez, Tania Gabaldón, Javier Hernández Laforet

**Affiliations:** Anesthesiology and Intensive Care Department, Consorcio Hospital General Universitario de Valencia, 46014 Valencia, Spain; tanya.gabaldon@gmail.com (T.G.); jaherla@hotmail.com (J.H.L.)

**Keywords:** air leak, pneumothorax, fistula, ECCO2R, extracorporeal CO_2_ removal, ultraprotective ventilation, ARDS

## Abstract

Extracorporeal CO_2_ removal (ECCO2R) is a therapeutic approach that allows protective ventilation in acute respiratory failure by preventing hypercapnia and subsequent acidosis. The main indications for ECCO2R in acute respiratory failure are COPD (chronic obstructive pulmonary disease) exacerbation, acute respiratory distress syndrome (ARDS) and other situations of asthmatics status. However, CO_2_ removal procedure is not extended to those ARDS patients presenting an air leak. Here, we report three cases of air leaks in patients with an ARDS that were successfully treated using a new ECCO2R device. Case 1 is a polytrauma patient that developed pneumothorax during the hospital stay, case 2 is a patient with a post-surgical bronchial fistula after an Ivor–Lewis esophagectomy, and case 3 is a COVID-19 patient who developed a spontaneous pneumothorax after being hospitalized for a prolonged time. ECCO2R allowed for protective ventilation mitigating VILI (ventilation-induced lung injury) and significantly improved hypercapnia and respiratory acidemia, allowing time for the native lung to heal. Although further investigation is needed, our observations seem to suggest that CO_2_ removal can be a safe and effective procedure in patients connected to mechanical ventilation with ARDS-associated air leaks.

## 1. Introduction

Pneumothorax and other forms of air leak are complications associated with acute respiratory distress syndrome (ARDS), the treatment of which constitutes a major challenge, even more if patients require ventilator support with invasive mechanical ventilation (IMV) as they are associated with long hospital stays and high morbidity [1,2]. Although it has proven its effectiveness in life-threatening conditions, IMV might be associated with significant adverse events [3,4]. Patients with barotrauma and air leaks represent a challenge in terms of ventilation. In these patients, in order not to exacerbate the air leaks, low level of PEEP and driving pressure are mandatory, but at the same time high tidal volumes might be required to allow for CO_2_ removal, given the low effective ventilation caused by the air leaks. Therefore, establishing a lung ultraprotective ventilation (LUV) to avoid the ventilator-induced lung injury (VILI) and to not exacerbate the air leaks is a crucial step for an effective clinical management of patients with ARDS and air leaks [5,6].

Lung ultraprotective ventilation can develop hypercapnia and acidosis and can be sustained in removing CO_2_ from blood stream by using a low extracorporeal blood flow and a high extracorporeal gas flow [5,6,7]. Extracorporeal CO_2_ removal (ECCO2R) was designed to allow ventilation in a protective manner without the consequences of hypercapnia and acidosis [8] allowing a reduction in the conventional ventilation requirements [9,10,11]. There is clinical evidence suggesting that ECCO2R is an effective and safe strategy in patients with ARDS [12,13,14,15,16,17]. However, most of these studies were performed with devices which had a smaller surface area of the gas exchange membrane.

We used a low flow ECCO2R technique, PrismaLung Plus+, a new version of the old PrismaLung membrane that optimized the surface according to the most common flows used in continuous renal replacement therapy (CRRT) monitors. As described by Karagiannidis and collaborators, at these flows, increasing exchange surface beyond 0.8 m^2^ would only offer a narrow gas exchange improvement [18,19]. Therefore, according to these observations, the new device uses a 0.8 m^2^ exchange membrane instead of 0.4 m^2^.

The purpose of this study is to present the clinical experience, in a real-life scenario, of the first three patients in Spain who were treated with this new device. Moreover, to the best of our knowledge, ECCO2R therapy was not previously applied to treat ARDS-associated air leak patients by allowing protective mechanical ventilation. Therefore, we report ARDS-associated air leak as a possible new indication for ECCO2R.

## 2. Materials and Methods

Design. Retrospective chart review of the first three patients who underwent treatment with the new ECCO2R device (PrismaLung+, Baxter Healthcare) was conducted. As it is a retrospective observational study with the revision of 3 medical charts, the Ethic Committee of Consorcio Hospital General Universitario de Valencia granted exemption of approval. ECCO2R therapy was started while all the patients were admitted into the intensive care unit (ICU). Families were informed of the rescue treatment and the benefits–risks ratio.

Any information that could lead to an individual being identified has been encrypted or removed, as appropriate, to guarantee their anonymity. The study protocol adhered to the tenets of the Declaration of Helsinki and the Good Clinical Practice/International Council for Harmonization Guidelines.

Study Participants. The current study included three patients, who were admitted to the ICU, with ARDS of different etiologies who needed invasive mechanical ventilation and whose common link was the presentation of persistent air leaks: case 1, polytrauma with persistent leak after pleural drainage because of pleural effusion; case 2, esophageal surgery [Ivor–Lewis esophagectomy] with iatrogenic bronchopleural and esophageal fistulous hole; case 3, coronavirus disease with spontaneous pneumothorax and persistent air leak after chest drainage. Cases 1 and 3 air leaks did not respond to conventional treatment (pleurodesis and blood patch) and ECCO2R therapy was indicated to allow the lowering of the tidal volume ventilation and to meanwhile allow the lung to heal. Case 2 was an iatrogenic air leak due to esophagus bronchopleural fistulous hole and ECCO2R therapy was used to allow ventilation until the definitive esophageal and endobronchial stent were placed and the hole sealed.

ECCO2R therapy. For the ECCO2R therapy in the three cases, we canulated the right femoral vein using the Seldinger technique and we used a high flow double-lumen catheter of 13F and 200 mm length; these are the same techniques and catheters that we use in our unit for continuous renal replacement therapy (TRR). Anticoagulation method used was heparin, just as we do for TRR (Table 1).

ECCO2R system. Prismalung+^®^ (Baxter Healthcare) is a CO_2_ removal device that uses a polymethylpentene hollow-fiber mats membrane, with a total surface area of the gas exchange membrane of 0.8 m^2^ [19]. This device is used with the PrisMax 2 platform.

CRRT. During ECCO2R therapy, patient #2 received continuous venovenous hemodiafiltration (CVVHDF) via the CRRT filter Oxiris (Baxter). Parameters were QB = 400 mL/min, QD = 1000 mL/h and QPFR = 100 mL/h.

Data recording. Demographic data collected included age, gender, primary admission diagnosis, cause of respiratory failure, and any known comorbidities. Ventilatory data included results of arterial blood gases, respiratory rate, tidal volume, driving pressure, minute ventilation, and positive end-expiratory pressure (PEEP) (Table 2).

Statistical analysis. Statistical analysis was performed with the MedCalc^®^ Statistical Software version 20.114 (MedCalc Software Ltd., Ostend, Belgium; https://www.medcalc.org; accessed on 11 November 2022). Due to the small number of patients, the statistical analyses were performed more with a purely descriptive purpose than analytical. Median and interquartile range (IqR), geometric mean, 95% confidence interval (95% CI), and number (percentage) were used as appropriated.

Comparisons between quantitative variables were performed using the two-way paired sample Student *t* test difference on log-transformed scale. Categorical variables were compared using a Fisher’s exact test.

## 3. Results

### 3.1. Baseline Results

Three patients with an ARDS were included in the analysis. The median age was 49 (25.8 to 56.5) years (range: 18 to 59 years). All the subjects were men and have been in the hospital for a long period before their conditions worsened and were candidates for undergoing ECCO2R. Reasons for this therapeutic approach were ARDS plus respiratory acidosis and hypercapnia with infection in two subjects and pH (acidosis) and refractory hypercapnia alone in one, and with persistent air leak in all of them. The main baseline demographic and clinical characteristics are shown in Table 1. In addition, patient #2 received continuous renal replacement therapy (CRRT) via the Oxiris filter (Baxter) in combination with ECCO2R therapy, as described in the Methods section.

### 3.2. ECCO2R Therapy and Ventilatory and Laboratory Parameters

In the three cases, we started ECCO2R therapy with a blood flow of 400–450 mL/h and gas flow O_2_ 15 L/min with ventilatory parameters adjustments and reevaluated the situation (ventilator and laboratory parameters) after 6 h of treatment as follows (ventilator and blood gas parameters before and after 6 h of ECCO2R are shown in Table 2).

The geometric mean (95% CI) arterial partial pressure of carbon dioxide (PaCO_2_) was significantly lowered from 92.7 (59.8 to 143.7) mmHg at baseline to 50.0 (45.9 to 54.4) mmHg after 6 h ECCO2R therapy (*p* = 0.0285; paired sample Student *t* test difference on log-transformed scale). Similarly, the baseline fraction of inspired oxygen (FiO_2_) was significantly reduced from 76.5% (63.2% to 92.7%) to 44.4% (26.6% to 70.1%) after LUV (*p* = 0.0182; paired sample Student *t* test difference on log-transformed scale).

Regarding pH, there was a significant increase from an average of 7.22 at baseline to 7.41 after LUV with ECCO2R therapy after 6 h (*p* = 0.0443; paired sample Student *t* test difference on log-transformed scale). Similarly, pretreatment tidal volume and tidal volume/weight significantly decreased from 458.1 mL (289.4 mL to 725.2 mL) and 6.31 mL/Kg (3.38 mL/Kg to 11.77 mL/Kg) to 351.2 mL (250.2 mL to 493.0 mL) and 4.82 mL/Kg (2.53 mL/Kg to 9.20 mL/Kg) post-LUV, respectively. Both were significant; *p* = 0.0384 and *p* = 0.0364, respectively. Although high PEEP levels are unfavorable for air leak, we maintained almost the same PEEP before and after ECCO2R treatment because of hypoxemia and elevated FiO_2_ (average PEEP of 7.7).

PEEP, arterial partial pressure of oxygen, respiratory frequency, and driving pressure did not show significant changes.

The median (IqR) time for ECCO2R therapy in LUV was 7.0 (4.8 to 9.3) days. We started ECCO2R weaning when we had a stable respiratory situation (same ventilatory and laboratory parameters as shown in table after treatment) and a decrease in air leak was by the 5–6th day by first lowering the blood flow and secondly by lowering the gas flow. By 5th day of ECCO2R treatment, we lowered blood flow to 300 mL/h and gas flow to 10 lpm during 24 h and then to 250 mL/h and 5 lpm in the next 24 h before stopping the therapy.

### 3.3. Radiological Findings

The radiological findings for the three cases are shown below. In all cases there are evident radiologic improvement before and after ECCO2R therapy. It should be acknowledged that the radiologic improvement could also be an effect of the improvement in the disease.

Figure 1 shows thoracic computerized axial tomography of case 1 (polytrauma with persistent air leak after thoracic drainage) before ECCO2R treatment at day 42 after ICU admission (A) and 10 days after ECCO2R treatment at day 55 of ICU admission (B). 

Radiological findings of case 2 (esophageal surgery [Ivor–Lewis esophagectomy] with iatrogenic bronchopleural and esophageal fistulous hole) are shown in Figure 2. The computer axial tomography (CAT) shows atelectasis of the right lower pulmonary lobe with left posterolateral passive atelectasis at admission (Figure 2A) and at discharge after endobronchial prothesis and ECCO2R therapy (Figure 2B)

Radiological findings of case 3 (coronavirus disease with spontaneous pneumothorax and persistent air leak after chest drainage) are shown in Figure 3 and Figure 4. 

## 4. Discussion

Critically ill patients requiring IMV have usually poor clinical outcomes, with the death rate of those patients being over 40% [20]. In addition, air leaks have been associated with acute lung injury and increased mortality in animal models of the acute respiratory distress syndrome [21] as well as in patients, where air leaks have been reported to increase the rate of mortality by 26% [22]. Before the introduction of the lung protection concept in ventilation, overall mortality for air leaks lasting for more than 24 h was reported to be 67% [23]. However, the adoption of protective ventilation procedures dramatically improved patient survival [24].

In order to avoid increased risk of hypercapnia and hypercapnic acidosis when using low tidal volumes and pressure [25], ECCO2R systems help by removing CO_2_ from the patient’s blood and thus avoiding hypercapnia and subsequent respiratory acidosis. Therefore, CO_2_ removal is currently considered as a valuable treatment option for patients with ARDS [12,13,14,15,16,17].

We present the clinical results of three critically ill patients whose commonality is the presence of air leaks. According to the Berlin definition, as patients did not present severe hypoxemia, extracorporeal membrane oxygenation (ECMO) was not the primary indication [26]. In addition, as barotrauma may be a manifestation of persistent bronchopleural air leak, or bronchopleural fistula (BPF) [23], we considered CO_2_ removal would be beneficial by allowing ultraprotective ventilation in these patients in whom high tidal volumes do not allow the lung to heal with benefits not only in lung mechanics but also with metabolic consequences, and in non-ECMO centers as well.

In our study, all three patients presented air leaks (two of them could be classified as persistent) from different etiologies: ARDS + infection in two cases and refractory hypercapnia in the other one. Air leaks, and particularly persistent air leaks, represent a challenging scenario in clinical practice. Unique circumstances in each case often necessitate an individualized approach based on patient and clinical factors [27]. Moreover, in critically ill patients, air leaks have to be addressed in the context of the patient’s overall condition [28], with concurrent infection and ARDS often complicating an already challenging clinical scenario [27].

Therapeutic management of air leaks in critically ill patients should start with the ventilator. Lung ultraprotective ventilatory strategies that limit airways pressures might be the treatment of choice in those patients, as they reduce the risk of further developing air leaks and favor its resolution once one has occurred [27]. According to Grotberg et al. [28], in the event of an air leak, both inspiratory pressure and PEEP need to be minimized. Otherwise, volume loss would increase (they presented a case report of a patient with a fistula in which it increased from 15% to 54% with the addition of a 15 cm H_2_O PEEP). Therefore, ECCO2R therapeutic approach may help to gain time while solving the air leakage.

In the current study, before starting with LUV, patients were in hypercapnia (mean pCO_2_: 92.7 mmHg), which led to acid–base disturbance (mean pH: 7.22). The data showed that the use of ECCO2R was associated with a significant reduction in PaCO_2_, FiO_2_, and tidal volume, while it also significantly reduced respiratory acidemia. According to our results, ECCO2R appeared to be an effective and safe strategy for treating critically ill patients with ARDS and air leaks (regardless of its cause). In two cases, leak volumes were lower after ECCO2R, although air leaks were not completely solved. Unfortunately, in the other case, leak volumes data were not collected.

We describe pneumothorax and pulmonary air leaks as possible new indications for ECCO2R. Patients with pneumothorax were excluded in several previous carbon dioxide removal clinical trials [13,14] due to the fact that pneumothorax was reported as one of the possible complications related to ECCO2R, specially to the cannulation process [13,29,30]. However, in our experience, no technique-related complications, such as catheter infection, hemorrhage or thrombosis were observed during the ECCO2R procedure; we used the same catheters and anticoagulation that we used with continuous renal replacement therapy.

The indication for using ECCO2R techniques is acute respiratory failure with acidosis and hypercapnia but not severe hypoxemia, thus allowing for lung protective ventilator settings and decreasing ventilator-induced lung injury. Here, we show the usefulness of using low flow carbon dioxide removal via ECCO2R (with a membrane surface of 0.8 m^2^) to enhance protective ventilation in air leak patients who required mechanical ventilation, so as to allow for enough time for the native lung to heal by reducing tidal volume, lung stress and strain by clearing enough CO_2,_ thereby reducing dependence on the injured native lung [31]. However, it is not still clear when this technique must be applied and which hypoxemic patients will benefit from them. There is a recent article by Dianti et al. [32] in which they conducted a secondary analysis of the REST trial. In this secondary analysis, they saw that the subgroup of patients with PaO_2_:FiO_2_ 110 mmHg or higher had a posterior probability for 98% benefit. Of note, in our case series, all three patients successfully recovered. Several limitations should be taken into consideration when interpreting the study results. The first one is the limited sample size. The second limitation was the statistics used; log transformation is a widely used method to address skewed data, however, its use is not free of controversy. There is not sufficient evidence to determine when to start ECCO2R techniques and to know for certain when they are effective. According to our experience, we suggest ECCO2R as a plausible effective measure for ARDS patients with air leaks, and in non-ECMO centers as well, but readers should be mindful about its limitations, particularly when interpreting the relevance of the analysis.

## 5. Conclusions

The results of the current study suggest that in patients with ARDS and air leaks (regardless of its etiology) ECCO2R allowed protective ventilation and significantly improved hypercapnia and respiratory acidemia. These results provide a new possible therapeutical option to manage ARDS-associated air leaks in mechanical ventilated patients which should be further explored in a larger sample of patients.

## Figures and Tables

**Figure 1 jpm-13-01081-f001:**
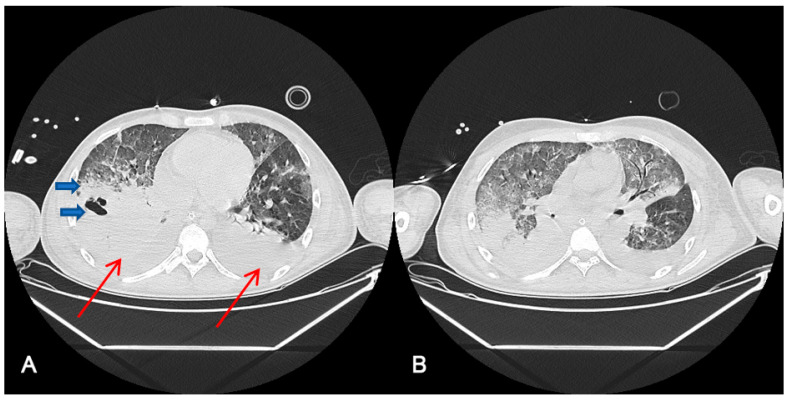
Radiological findings of case 1 ((**A**). Day 42 after ICU admission, before ECCO2R treatment (**B**). After 10 days ECCO2R treatment. Day 55 after ICU admission). Case #1: Thoracic computerized axial tomography that shows bilateral alveolar condensations affecting all lung lobes indicating a significant radiological worsening. Bilateral pleural effusion of up to 22 mm in the right hemithorax and 30 mm in the left one (red arrows). An intraparenchymal bubble (blue arrow) of approximately 30 by 26 mm is observed in the anterior basal segment of the right lower lobe, which does not allow abscessification to be ruled out (**A**). After ECCO2R therapy (**B**), it should be acknowledged that the radiologic improvement could also be an effect of the improvement in the disease.

**Figure 2 jpm-13-01081-f002:**
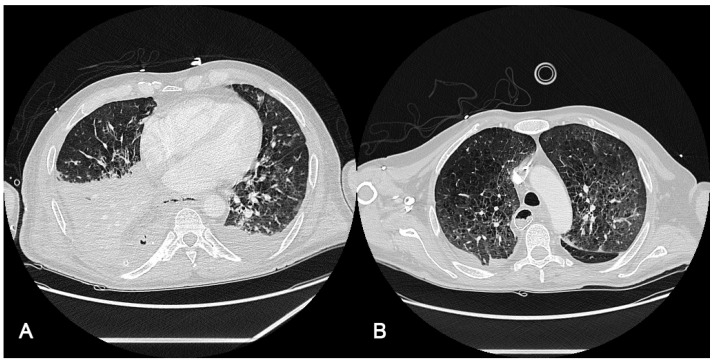
Radiological findings of case 2. Case #2: (**A**): The computed axial tomography (CAT) scan shows an atelectasis of the right lower pulmonary lobe with discrete bilateral pleural effusion, predominantly right, partially encapsulated. Mild left pleural effusion with left posterolateral passive atelectasis. (**B**): CAT at discharge, 20 days after ICU admission, after endobronchial prothesis and ECCO2R therapy. It should be taken into account that radiological findings improvement may be due to pathology resolution (fistulous hole).

**Figure 3 jpm-13-01081-f003:**
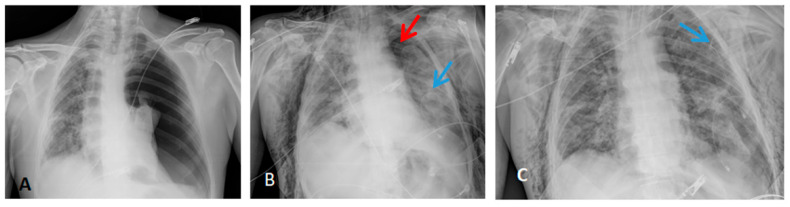
Thorax X-ray findings of case 3. (**A**) Massive left pneumothorax. (**B**) Apical left pneumothorax (red arrow) with partial reexpansion after pleural drainage (blue arrow). (**C**) Incomplete lung reexpansion and air leak after second pleural drainage (blue arrow).

**Figure 4 jpm-13-01081-f004:**
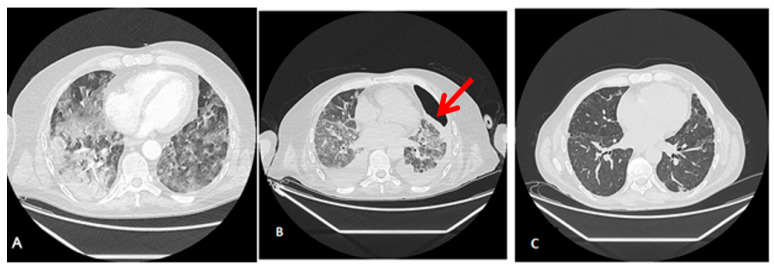
Radiological findings of case 3. Case #3: (**A**) At ICU Admission: Pulmonary thromboembolism with signs of pulmonary hypertension. Bilateral alveolar interstitial infiltrate due to pneumonitis secondary to SARS-Cov-2 infection. (**B**) Anterior pneumothorax (red arrow) with partial atelectasis and bilateral alveolar infiltrate with lower density and extension. (**C**) At discharge, 3 months after admission, a clinically significant improvement in the alveolar-interstitial infiltrates and resolution of the pneumothorax are evident.

**Table 1 jpm-13-01081-t001:** Main demographic and clinical baseline characteristics at admission.

	Case 1	Case 2	Case 3	Overall
Reason for admission	Polytrauma	Esophagectomy	SARS-CoV-2 Delta	N.A.
Reason for ECCOR2	ARDS+infection	Refractory hypercapnia	ARDS+infection	N.A.
Age, years Median (IqR) Range	18	59	49	49 (25.8 to 56.5) 18 to 59
Sex	Male	Male	Male	Male (100.0)
Weight (Kg) Median (IqR) Range	60	80	80	80 (65 to 80) 60 to 80
Comorbidities	No	ADC **	No	N.A.
Inotropes	No	No	No	N.A.
Nitric Oxide	6 ppm	No	8 ppm	N.A.
Air leak	Yes *	>250 mL/min	300 mL/min	N.A.
Air leak etiology	Secondary Spontaneous Pneumothorax	Iatrogenic pneumothorax due to surgery	Grade 3 persistent secondary spontaneous pneumothorax (coalesced bubbles)	N.A.
SOFA at Admission Median (IqR) Range	5	3	3	3.0 (3.0 to 4.5) 3 to 5
SOFA at ECCO2R start Median (IqR) Range	6	12	9	9.0 (6.8 to 11.3) 6 to 12
ECCO2R treatment, days Median (IqR) Range	10	4	7	7.0 (4.8 to 9.3) 4 to 10
Anticoagulation Heparin LMWH (bemiparine)	10–13 UI/Kg/h 2500 UI (prophylaxis)	7 UI/Kg/h 3500 UI (prophylaxis)	5–10 UI/Kg/h *** After 48 h, 5000 UI/24 h	N.A.
CRRT	No	CVVHDF—Oxiris	No	N.A.
Main infection	Candiduria, Pseudomona	No infection detected	Staphylococcus, Klebsiella, Pseudomona, C. auris, Elizabethkingia meningoseptica	N.A.

* Not specified. ** ADC of lower third of esophagus cT3N2Mx *** Before bemiparine was added, between 24 h and 48 h, heparin rate was raised to 12.5 UI/Kg/h. ECCO2R: Extracorporeal CO_2_ removal; ARDS: Acute respiratory distress syndrome; ASDC: Adenocarcinoma; IqR: Interquartile range; N.A.: Not applicable; CRRT: Continuous renal replacement therapy.

**Table 2 jpm-13-01081-t002:** A comparison of the ventilatory and laboratory parameters before and after ultraprotective ventilation.

	Before				After 6 h ECCO2R				*p* ^a^
	Case 1	Case 2	Case 3	Geometric Mean (95% CI)	Case 1	Case 2	Case 3	Geometric Mean (95% CI)	
pH	7.2	7.14	7.33	7.22 (6.99 to 7.47)	7.46	7.32	7.45	7.41 (7.22 to 7.61)	0.0443
pCO_2_, mmHg	112	90	79	92.7 (59.8 to 143.7)	49	52	49	50.0 (45.9 to 54.4)	0.0285
pO_2_, mmHg	73	94	94	86.4 (60.1 to 124.2)	161	83	118	116.4 (51.1 to 265.2)	0.3800
TV (mL)	460	550	380	458.1 (289.4 to 725.2)	380	380	300	351.2 (250.2 to 493.0)	0.0384
TV/weight(mL/kg)	7.67	6.88	4.75	6.31 (3.38 to 11.77)	6.3	4.75	3.75	4.82 (2.53 to 9.20)	0.0364
RF, pm	20	17	18	18.3 (14.9 to 22.5)	12	15	18	14.8 (8.9 to 24.5)	0.3018
PEEP, mmHg	8	8	7	7.7 (6.3 to 9.3)	10	6	7	7.5 (3.9 to 14.4)	0.8977
Leak, mL	N.A.	250	300	N.A.	N.A.	180	100	N.A.	N.A.
DP, mmHg	24	15	10	15.3 (5.2 to 45.5)	8	15	7	9.4 (3.4 to 25.9)	0.2726
FiO_2_, %	80	70	80	76.5 (63.2 to 92.7)	50	35	50	44.4 (26.6 to 70.1)	0.0182

^a^ Paired sample Student *t* test difference on log-transformed scale. I: Confidence interval; pCO_2_: Partial pressure of carbon dioxide; pO_2_: Partial pressure of oxygen; TV: Tidal volume; RF: Respiratory frequency; PEEP: Positive end-expiratory pressure; DP: Driving pressure; FiO_2_: Fraction of inspired oxygen; N.A.: Not available.

## Data Availability

No new data were created or analyzed in this study. Data sharing is not applicable to this article.

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
