# Peer review of "Ultraprotective Ventilation via ECCO2R in Three Patients Presenting an Air Leak: Is ECCO2R Effective?"

_jpm, 2023, doi:10.3390/jpm13071081_

Round 1

Reviewer 1 Report

This manuscript presents a case report aimed to present the clinical experience of the new ECCO2R device (PrismaLung+, Baxter Healthcare) in a real-life scenario, while three ARDS-associated air leak patients in Spain were treated with this new device. The study included three patients, who were admitted to ICU, with ARDS of different etiologies (Polytrauma, esophageal surgery [Ivor-Lewis esophagectomy], and coronavirus disease [COVID-19]), whose common link was the presentation of air leaks. The ECCO2R system assembled by Prismalung+® is a CO2 removal device that used a polymethylpentene hollow-fiber mats membrane, with a total surface area of the gas exchange membrane of 0.8m2. The results showed ECCO2R therapeutic approach might help to gain time while solving the air leakage. However, there were many key comments that had not been clearly described in this article.

1.The ECCO2R therapy was a controversial low flow Extracorporeal Life Support treatment. Although the study added some new information that may be useful for caring for this population of ARDS-associated air leak patients, there was no significant promotion of its application due to individual differences among the three patients.

2.The article did not disclose important information such as the type of the patient's intravascular catheter, changes in blood flow during treatment, and carbon dioxide clearance efficiency.

3.Although the ventilator parameter settings were elaborated, the specific time for correcting the patient's internal environment (how long after ECCO2R treatment for remission) was not clearly defined, and the weight of ECCO2R in patient remission cannot be fully evaluated. We were not sure if patients could improve their condition without ECCO2R treatment.

4.     I noticed that the PEEP levels were high in three patients before and after ECCO2R. In Case 1, the PEEP level increased from 8 cmH2O to 10 cmH2O after ECCO2R, which is unfavorable for air leak. Why were the ventilator parameters adjusted in this way?

In terms of language expression, it is suggested to invite a native speaker to help revise the article.

Reviewer 2 Report

Thanks for the opportunity to revise this interesting paper. In their work Ferrer and colleagues propose the use of ECCO2R in ARDS patients with air leaks.
The manuscript is interesting, but there is room for improvement.
Please, find below some comments:

1.       In the introduction I would suggest to stress a bit more the difficulties in ventilating patients with barotrauma/air leaks. In this patients often low level of peep and driving pressure are mandatory (to not exacerbate the air leaks), but at the same time high tidal volumes might be required to allow for CO2 removal, given the low effective ventilation (caused by the air leaks).
The authors could emphasize this concept (which is why their work is novel and makes a lot of sense).
To keep the length of the introduction reasonable, they could shorten the initial part on the general description of VILI and mechanical ventilation, as it is well known among the audience interested in the ECCO2R.

2.       Matherials and methods: Ok

3.       Results, baseline: in the text it could be helpful to briefly describe the lung mechanics and the blood gases  (at least the paco2 and the ph) of the patients just before the ECCO2R. I understand that you already did that in the table 2, but the reader should be able to follow the text without referring too much to the tables.

4.       The authors should mention if the patients were treated with a chest drainage (I assume they were).

5.       The radiology is very interesting and appealing. However, I have the feeling it could be improved:

a.       You could add some arrows or markers to help the reader visualize the point of interests (air bubble, fistula, pneumothorax.

b.       The exact timing of image A and B should be reported (especially if image A was taken at day 1 and image B at day 15).

c.       The improvement in image B could be only an effect of the improvement in the disease, without any specific effect of the ECCO2R. This should be acknowledged.

6.       You should clarify what do you mean as “before and after ECCO2R”, regarding lung mechanics. Also, it could be helpful to explain how long it took to reach the steady state in PaCO2 and therefore ventilatory settings.

7.       In the discussion and conclusions, due to the nature of your data, I think that you are too strongly suggesting that ECCO2R can be an effective measure for ARDS patients with air leaks.
I would suggest modulating a little bit the strength of your message in the light of you results, that although interesting have some limitations.

The fluency of the paragraph could be improved, but overall good. Moderate revision of the English could increase the readability of the manuscript.
